# Overcoming gender inequality for climate resilient development

Marina Andrijevic [1,2✉], Jesus Crespo Cuaresma[3,4,5,6], Tabea Lissner[2], Adelle Thomas [1,7] & Carl-Friedrich Schleussner [1,2]

Gender inequalities are reflected in differential vulnerability, and exposure to the hazards posed by climate change and addressing them is key to increase the adaptive capacities of societies. We provide trajectories of the Gender Inequality Index (GII) alongside the Shared-Socioeconomic Pathways (SSPs), a scenario framework widely used in climate science. Here we find that rapid improvements in gender inequality are possible under a sustainable development scenario already in the near-term. The share of girls growing up in countries with the highest gender inequality could be reduced to about 24% in 2030 compared to about 70% today. Largely overcoming gender inequality as assessed in the GII would be within reach by mid-century. Under less optimistic scenarios, gender inequality may persist throughout the 21st century. Our results highlight the importance of incorporating gender in scenarios assessing future climate impacts and underscore the relevance of addressing gender inequalities in policies aiming to foster climate resilient development.

[1] IRI THESys, Humboldt University, 10117 Berlin, Germany. [2] Climate Analytics, 10969 Berlin, Germany. [3] Vienna University of Economics and Business (WU), 1020 Vienna, Austria. [4] Wittgenstein Centre for Demography and Global Human Capital (IIASA, OeAW, University of Vienna), International Institute for Applied Systems, 2361 Laxenburg, Austria. [5] Austrian Institute of Economic Research (WIFO), 1030 Vienna, Austria. [6] CESifo, 81679 Munich, Germany. [7] University of the Bahamas, Nassau 4912, Bahamas. ✉email: marina.andrijevic@hu-berlin.de

Differential risks to climate change impacts are shaped by variations in vulnerability and exposure within and across societies. Together with their biophysical determinants, vulnerability, and exposure are products of unevenly distributed socioeconomic development and multidimensional inequality[1]. Inequalities are reflected in income and wealth, which remain central subjects of socioeconomic research, but also in gender, education, racial, and ethnic profiles[2]. Socially marginalized groups are often affected by the interplay of these different dimensions and are more vulnerable to the impacts of climate change.

A growing body of literature points at the facets of differential vulnerability and exposure to the impacts of climate change across genders, stressing that women are not inherently more at risk, but that intersections between gender, power dynamics, socio-economic structures, and societal expectations result in climate impacts being experienced very differently by women[3]. Research has also highlighted missed opportunities for action when women's agency in policy and decision making is not fully seized[4]. In our contribution, we focus on the role of gender inequality, which despite its prominence as a cross-cutting theme in the sustainable development discourse, lacks concrete operationalizations in the analysis of future impacts of climate change and the extent to which these can still be avoided[5].

Current and future damages of climate change are tied to the ability with which affected regions and populations adapt to changing conditions. In the risk framework of the Fifth Assessment Report (AR5) of the United Nations Intergovernmental Panel on Climate Change (IPCC), vulnerability to climate change impacts is inextricably linked to adaptive capacity, which is defined as "the ability of systems, institutions, humans, and other organisms to adjust to potential damage, to take advantage of opportunities, or to respond to consequences"[6]. Adaptive capacity, in turn, hinges on a range of socioeconomic factors, gender inequality playing one of the central roles, particularly in areas most vulnerable to climate change. The linkages between gender inequality and adaptive capacity range from uneven access to resources, to cultural norms and entrenched social structures[7,8].

Accounting for gender inequality and its possible future trajectories in the assessment of the pathways of adaptive capacity adds another layer to the identification of societal climate impact hotspots—areas where expected biophysical impacts intersect with socioeconomic vulnerability[9,10]. In this paper, we present an extension of the set of socioeconomic scenarios—the Shared Socioeconomic Pathways (SSPs)[11]—with an indicator of gender inequality, the Gender Inequality Index (GII)[12] of the United Nations Development Programme (UNDP). The SSPs are a widely used toolkit in climate change research and provide a basis for the operationalization of indicators of gender inequality in integrated assessments.

The GII used here to reflect gender inequality consists of three dimensions: health (maternal mortality ratio and adolescent birth rates), educational and political empowerment (male to female ratio in parliamentary seats and secondary education) and participation in the labor market (male to female ratio in labor force participation rates, see the "Methods" section for additional details on the indicator)[12]. We collected the individual components from their respective original sources and reconstructed the index following the approach laid out in the Technical Notes of the Human Development Report[12]. This reconstruction produced more complete time series than those available hitherto (see Supplementary Fig. 1). The index ranges from 0 to 1, with higher values reflecting higher levels of inequality between men and women.

The multi-faceted nature of gender inequality at all levels of socio-economic development makes aggregation into indicator a complex exercise. Unsurprisingly, most indicators (including the GII), face justified criticism[13,14] (see the "Methods" section for an extended discussion). We consider the dimensions covered in the GII to describe necessary conditions of gender inequality, while acknowledging that they are not sufficient to characterize gender inequality across all the dimensions that contribute to it. In the light of these caveats, overcoming the inequality dimensions covered in the GII does not automatically mean that universal gender equality is achieved, and we do not assert that any country in the world can claim to have achieved full gender equality to date or in the near future. It is important to keep these limitations in mind when interpreting the results.

The ramifications of gender inequality for addressing climate change can be regarded through two lenses: women's differential vulnerability and adaptive capacity; and the role of women in mitigation and adaptation actions. To illustrate the importance of accounting for gender inequality in both adaptation and mitigation of climate change, we correlate the GII with an adaptation-relevant and a mitigation-relevant metrics (compare Fig. 1).

Previous research shows that the gender-differentiated vulnerability to climate change is most pronounced in agriculture[15,16] and water[17,18] sectors, natural disasters[19], reproductive health[20], mental health, and well-being[21]. We use a broad measure of climate change vulnerability of the Notre Dame Global Adaptation Index (ND-GAIN)[22], a widely used summary measure of a country's vulnerability to climate change and its readiness to improve resilience (for more applications, see refs. [23–25]). Figure 1a depicts the correlation between the GII and the ND-GAIN vulnerability indicator (consisting of six life-supporting sectors: food, water, health, ecosystem services, human habitat, and infrastructure), and depicts a strong positive relationship between the two variables.

At the same time, a strand of research suggests that women's representation in politics leads to more stringent climate action[26,27], thus making a case for consideration of mainstreaming gender equality in mitigation. More broadly, female participation in decision-making is closely linked to various facets of socioeconomic progress: from higher spending on health and education to better quality of institutions, democracy and higher economic growth[26,28–30]. Following a recent approach[26], in Fig. 1b we correlate the GII with the Climate Laws, Institutions and Measures Index (CLIMI)[31], a measure of climate change mitigation policies set by countries (for more applications, see refs. [32,33]). The correlation of the two indices suggests that low levels of gender inequality tend to occur in parallel to high levels of climate action, which corroborates previous research[26].

## Results and discussion

While the importance of rapid and stringent mitigation cannot be overemphasized, and recent research insights provide indications that gender equality facilitates climate action, here we focus on the importance of gender equality for adaptive capacity and vulnerability to climate change. To this end, we expand the scenario space of the Shared Socioeconomic Pathways (SSPs), with the intention of improving the understanding of adaptation challenges under different socio-economic conditions. The SSPs are scenarios that explore a range of possible futures that illustrate how socio-economic conditions might change over the next century and what implications these conditions may have for climate change adaptation and mitigation. SSPs quantify five different narratives of socio-economic futures to operationalize them for climate change research[11]—they are a widely used tool in climate research community, indispensable for integrated assessments of the dynamics between socioeconomic and climate change variables, and are also the scenario framework used in the Sixth Assessment report of the IPCC.

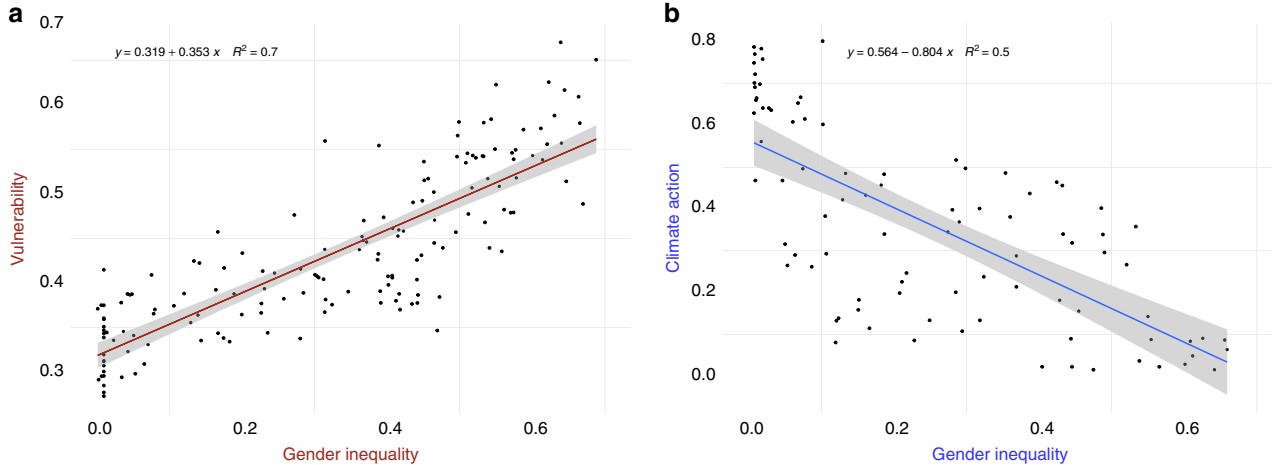

**Fig. 1 Gender Inequality Index (GII)—correlation with vulnerability and climate actions. a** GII vs. vulnerability component of the ND-GAIN index (country-level estimates for 2017). **b** GII (country-level average 2005–2010) vs. CLIMI (countries' communications of climate policies between 2005 and 2010).

| Table 1 Representation of gender inequality in SSP storylines[11]. | | | | |
| --- | --- | --- | --- | --- |
| **SSP1** | **SSP2** | **SSP3** | **SSP4** | **SSP5** |
| Low | Medium | High | High in LICs, low in HICs | Low |
| HIC/LIC: High/Low Income Countries. | | | | |

SSP1, the 'sustainability' scenario, is characterized by low challenges to mitigation and adaptation, a result of increased investments in education, health, renewable energy sources and declining inequalities between and within countries, thus limiting impacts and increasing adaptive capacity. SSP2, the 'middle of the road' scenario, maintains premediated challenges to adaptation and mitigation, and is a pathway of uneven and slower socio-economic progress, compatible with the continuation of historical trends. SSP3 is characterized by high challenges to both mitigation and adaptation, which are a product of a growing divergence between economies, weak international cooperation and increase in internal and international conflicts. SSP4, the scenario of 'inequality', leads to low challenges for mitigation, due to technological advancements in high income countries, but high challenges for adaptation, because of an unequal distribution of advancements and resources across countries. Finally, SSP5 is similar to SSP1 in the fast socioeconomic progress on all fronts, but with the major difference of the progress being powered by fossil fuels, which produces substantially higher emissions and resulting climate impacts.

So far, the SSPs storylines have been quantified in future trajectories of income[34,35], population[36], education[36], urbanization[37], the Human Development Index[38], inequality[39], and governance[40]. Gender inequality is qualitatively featured in the scenarios' storylines focusing on the demographic and human development elements (see Table 1), and is to a certain extent reflected in the measures of discrepancies in educational attainment between men and women in the population projections by age and sex[36]. Our contribution provides projections of gender inequality, as quantified by the GII, which are compatible with the SSP scenarios described above and thus provide a new dimension to the assessment of potential future climate change adaptation pathways.

To achieve an internally consistent extension of the SSPs, we use the existing indicators under the SSP framework to analyze past trends and project future dynamics of gender equality. Our results indicate that past trends in the GII can be robustly explained by the dynamics of GDP per capita, population with post-secondary education and the gender gap in mean years of schooling after controlling for country-specific equilibria and global trends (see "Methods" for regression results and Supplementary Material for a sensitivity analysis). As is the case within the methodological framework of the SSPs, the projections of the GII are not to be interpreted as predictions, but as quantifications of narrative-driven scenarios.

Our projection exercise shows that major improvements in terms of overcoming gender inequality are achieved worldwide by mid-century under the SSP 1 scenario (Fig. 2c). Significant improvements happen following the SSP2 (Fig. 2d) pathway, though with notable exceptions in the most vulnerable parts of the world. In the SSP3 world (Fig. 2e), however, only marginal progress is made in parts of Latin America, while in Sub Saharan Africa gender inequality is projected to deteriorate (compare Fig. 2e).

Given the central role that gender equality has for adaptive capacity, the future outlook concerning how well a country or a region can cope with the impacts of climate change can be very different depending on the scenario of socio-economic development. Across all world regions, improvements in gender equality in inclusive high-development pathways (SSP1, 5) are most pronounced in the near-term until mid-century. Note that the trajectories for SSPs 1 and 5 largely overlap due to similar levels of the underlying dimensions that gender inequality is a function of (education, GDP and gender gap in mean years of schooling). The summary of regional levels of gender inequality in Fig. 3 reflects the severity of the difference in levels of the GII, and the importance of near-term improvements for less well-off regions. As it is the case for other indicators of socio-economic development[38,40], the rates of improvement in the GII towards gender equality are highest up to 2050 in these scenarios. Less optimistic development pathways show a linear continuation of current trends or even a slow-down. Note that, by design, the SSPs do not allow for a systematic long-run deterioration of socio-economic indicators.

In the wider context of sustainable development—still inextricably linked to the climate change problem—the gender dimension is a crucial policy component, including as a standalone item under the Sustainable Development Goals (SDGs) of the United Nations' 2030 agenda. SDG 5 strives to "achieve

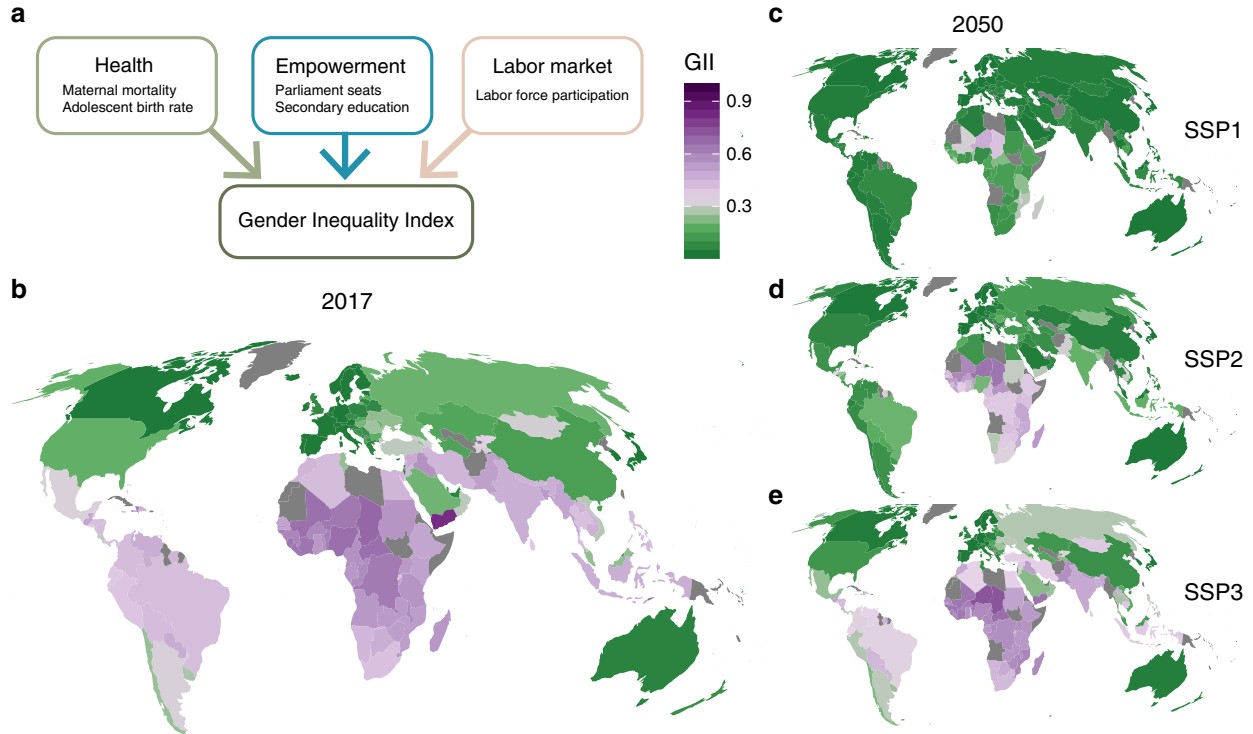

**Fig. 2 Present values and future projections of the Gender Inequality Index (GII). a** Components of the GII. **b** Values of the GII in 2017. **c–e** Projections of the GII for the year 2050, for **c**, SSP1 ('sustainability'), **d** SSP2 ('middle of the road') and (**e**) SSP3 ('a rocky road').

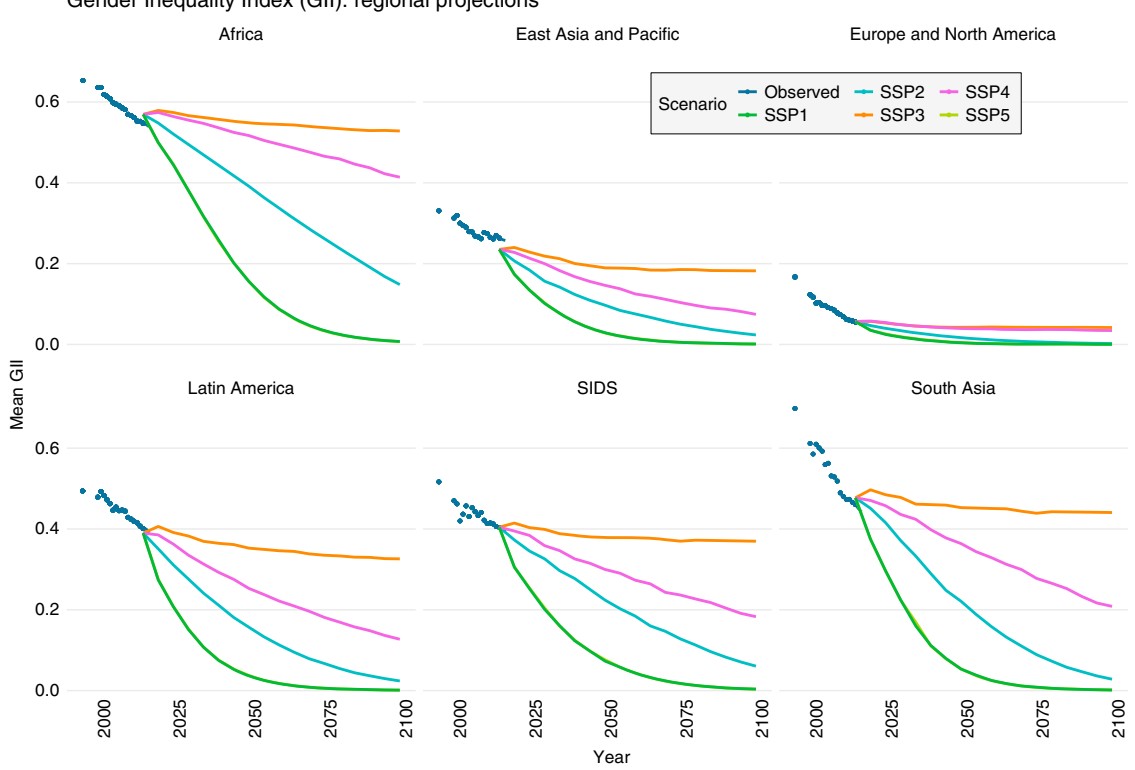

**Fig. 3 Evolution of the GII over the 21st century—regional outlook.** Historical values of the GII index and projections over five SSP scenarios, averaged by world region.

gender equality and empower all women and girls"[41], and the progress towards the multiple goals under SDG 5 is tracked with a set of individual indicators. The Gender Inequality Index presented here is a more holistic measure than the specific indicators

used in monitoring SDG 5. With its dimensions related to reproductive health and decision-making, as well as political and employment participation, it relates to underlying structural issues determining gender inequality[42]. As such, the GII and its

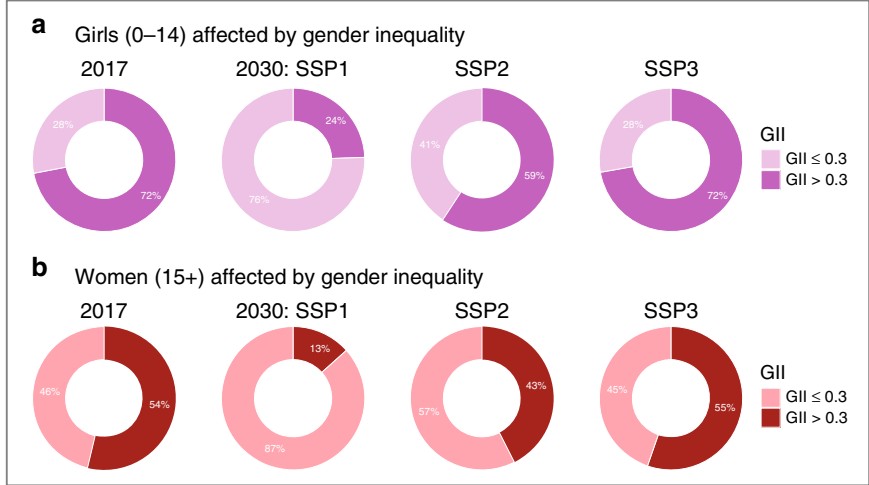

**Fig. 4 Share of women affected by gender inequality globally in 2020 and 2030.** GII values for 2017 and projections for 2030 are divided in two groups. The division is based is based on the present-day range of GII in the OECD countries (0.001–0.312), which splits the countries in GII ≤ 0.3 and GII > 0.3. The GII estimates are coupled with population projections disaggregated by female population projections for two broad age groups: (**a**), 0–14 years and (**b**), older than 15.

projections can be a useful tool to assess how the very basic conditions for making progress on SDG 5 vary in different socioeconomic futures.

Many of the countries experiencing high levels of gender inequality are in the mid-stages of the demographic transition[43], implying that their populations are expected to substantially grow in the next decades. Such a demographic development exposes young women to slow improvements in health, as well as to unequal opportunities in education and employment. Given the relatively high life expectancy of women born today, the level of gender inequality they are exposed to in the next decade will affect a cohort who will shape most of the 21st century. Figure 4 illustrates the opportunities for near-term improvements of gender inequality: already in 2030, the fraction of young girls growing up in environments of lower gender inequality (the present-day range of the GII in OECD countries) can be more than 2.5 larger in a pathway such as SSP1, where rates of population growth slow down and socioeconomic progress speeds up. On the other hand, scenario SSP3 virtually retains the present global distribution of our gender inequality indicator, due to faster population growth and slower and uneven socioeconomic development up to 2030. This underscores how rapid improvements towards achieving gender equality in the near-term would be possible, in line with the goals of the SDG 5. Note that for reasons of brevity we here show only scenarios 1–3, which encompass the full range of the five scenarios, and exhibit large differences between each other.

Our analysis outlines potential future gender inequality pathways under different scenarios of socio-economic development outlined in the SSPs. Our projections show that SSP1 results in major improvements in gender equality on a global scale while SSP2 shows some significant improvements but with notable exceptions in the most vulnerable regions, including Africa. In contrast, in the SSP3 world, gender inequality at the global level is either only marginally reduced or, in some cases, intensified. We show how such pathways may achieve concrete near-term improvements in the gender inequality environment for girls in the coming decade or may contribute to maintaining the status quo. The environments of gender inequality have significant implications for the growing global population, whose actions affect achievement of the SDGs. As a crucial component of adaptive capacity, gender inequality also plays a decisive role in allowing populations to adapt to increasing climate impacts.

Overcoming gender inequality is a cornerstone of climate resilient development—and improvements may have far-reaching benefits for adaptation and mitigation alike. Achieving climate resilience has to be designed in a way that not only prevents further erosion of gender equality, but actively works towards it, thereby reducing vulnerability and providing an empowering environment for strengthening women's agency.

## Methods

**Data**. *Gender Inequality Index (GII)*: the analysis in this paper is based on the GII[12], produced by the United Nations Development Programme. It integrates measures of reproductive health (maternal mortality ratio, adolescent birth rate), empowerment (secondary education, parliamentary seats), and labor market outputs (labor force participation rate).

The GII has been criticized on several grounds[13,44], with key issues relating to its functional form (which is asserted to be unnecessarily complex and difficult to interpret); the health dimension of the index variables not having a male equivalent (unlike the dimensions of economic, political and labor market metrics); and the potential penalization of poor countries owing to the possibility that poor reproductive health is a result of general poverty rather than gender inequality. Attempts have been made to simplify the index and make its interpretation more intuitive, though no clear consensus on how exactly the adapted indicator should look like has been reached, and to our best knowledge, the UNDP has not made any amends to the index so far.

The criticism about the penalization of less developed countries is concerned with the indicator's health dimensions (i.e., maternal mortality and adolescent birth rates), which could be caused by poverty rather than gender inequality, thereby obscuring the implications of this dimension. The very rationale behind accounting for maternal mortality and adolescent birth rate as a dimension of gendered health inequality stems from the fact that poor maternal health sets women back uniquely, irrespective of the reason and without an equivalent risk for men, and as such arguably contributes to gender inequality. Reducing maternal mortality and adolescent pregnancy are also among the targets of the Sustainable Development Goal 5 on gender equality[41]. In addition, recent applications found that the GII explains variance in child malnutrition and mortality in low and middle-income countries with similar income levels[45], implying that there the index does provide information on the variation of gender inequality across countries beyond that contained in GDP per capita differences. Finally, the fact that reproductive health is strongly affected by climate change impacts such as extreme heat is particularly relevant for the projection exercise presented here, and as such merits consideration as an own standing dimension of climate adaptation[46].

Further support for the GII's reflection of a broader understanding of gender inequality can be found in studies where it is found to correlate with other manifestations of gender inequality that go beyond what is included in the calculation of the index, such as the suicide gender ratio[47], adolescent dating violence[48], and intimate partner violence[49].

**Alternative indicators of gender equality**. Alternative indicators available in the literature incorporate different aspects of gender inequality. In the following, three other indicators will be introduced and examined in relation to the GII.

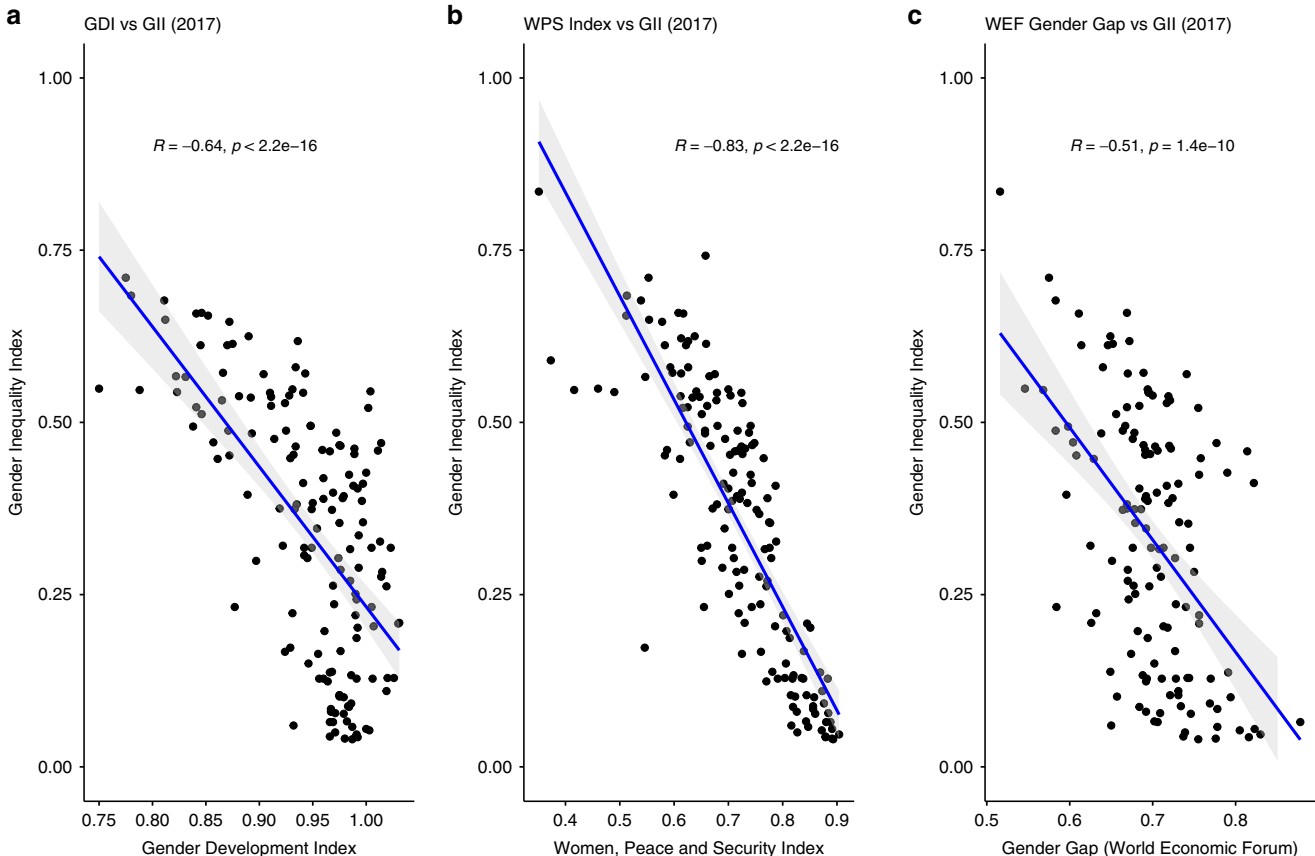

**Fig. 5 Comparison of the GII and other indices of gender equality.** Correlation coefficient (*R*) and the statistical significance (p) are provided for the relationship between GII and (**a**) Gender Development Index, **b** Women, Peace, and Security Index, and (**c**) Gender Gap index.

*Gender Development Index (GDI):* The GDI[12] is designed within the Human Development Reports provided by the United Nations Development Programme. Similarly to the Gender Inequality Index, it accounts for metrics of health, education and economic empowerment. The economic component of the index is difficult to reconstruct due to the scarcity of data on the wage gap between women and men, which is necessary for the calculation of the overall index. In addition, variation between countries is not as large as in the GII index, and the GDI does not capture basic metrics such as maternal and adolescent health, which are relevant for climate change vulnerability. The correlation of the GDI with the GII is depicted in Fig. 5a.

*Women, Peace and Security Index (WPS):* The WPS[50] is provided by the Georgetown Institute for Women, Peace, and Security and index captures three dimensions: inclusion (economic, social, political), justice (formal laws and informal discrimination) and security (violence, safety). Even though this index incorporates dimensions of high relevance for climate change-related vulnerability (particularly violence), it is only available at two points in time and is therefore suboptimal for the estimation of the historical response function that underpins our analysis. However, it is highly correlated to the GII used in this paper (see Fig. 5b).

*Global Gender Gap Index (GGI):* produced by the World Economic Forum, the GGI[51] incorporates four dimensions: economic participation, educational attainment, health and survival and political empowerment. The dimensions are represented by 14 different indicators. Compared to the GII used in this analysis, the GGI contains similar dimensions and there are overlaps among the underlying indicators to the GII used in this analysis, while the major difference is in the health component, where the GII considers maternal mortality and adolescent pregnancy, while the GGG takes into account life expectancy. Similarly to other indices, the time series of GGI is shorter than that of the GII. The GGI has the lowest (albeit statistically significant) correlation coefficient with the GII (Fig. 5c).

**Gender equality indicators and climate adaptation.** Compared to other commonly used indicators including the Gender Development Index[12], the Gender Empowerment Measure[51], and the Women, Peace and Security Index[50], we find that the GII is particularly indicative of hindered adaptive capacity in many climate-vulnerable countries, since its dimensions (such as maternal health, participation in economic and political life) point at the very basic disempowerment of women that directly reduces their capacity to adapt to climate change. The GII is also more holistic in its economic dimension, by considering education and labor

force participation rather than income, since the data on gender gap in earned income tends to be problematic[52]. In addition, the construction of the GII precludes the different dimensions of the indicator from compensating for each other (i.e., poor performance in one dimension cannot be compensated for with higher performance in another dimension in GII). While this is beyond the scope of this paper, application of our analytical framework to different indicators of gender inequality and analyzing the effect of the choice of the indicator on projections could be a fruitful research avenue.

Following the approach laid out in the Technical Notes of the Human Development Report (2018), we reconstructed the GII with the same underlying indicators, with the aim of obtaining more complete time series than those available hitherto. The data are available for majority of countries and can be reconstructed back to 1995 (see Supplementary Fig. 1). To capitalize on data availability and completeness, we use the same source indicators except for the education component, which we source from the Wittgenstein Centre for Demography and Global Human Capital[36] for better consistency with the projections that follow in the second stage of the analysis. The calculation of inequality uses an association-sensitive method, with geometric means of the three dimensions calculated for each gender separately, and then aggregated across genders using a harmonic mean. For comparison of the reconstructed GII and the data provided through the UNDP website, see Supplementary Fig. 1. Data analysis and projections were done using R software version 1.3.1073.

**Model.** To analyze the relationship between gender inequality and other socio-economic dimensions, we use a simple econometric model that expresses the GII as a function of GDP per capita, the share of population with higher education and the difference in mean years of schooling between men and women, and accounts for country-specific time-invariant characteristics using fixed effects. The model is aimed at replicating long-run dynamics in GII, with the theoretical underpinning that trends in socioeconomic variables correlate with the changes observed in gender inequality over long periods of time. From an econometric point of view, it can be considered a cointegration relationship posing common trends in gender inequality, income and human capital indicators around a country-specific equilibrium.

Prior to the analysis, the GII is transformed to account for the bounded nature of the index, which is defined between 0 and 1. The variable used in the panel regression models is given by $\mathrm{GII}^* = \log\left(\frac{\mathrm{GII}_{i,t}}{1-\mathrm{GII}_{i,t}}\right)$, where $\mathrm{GII}_{i,t}$ is the original

Gender Inequality Index for country i in period t. Our basic specification is given by:

$$\text{GII}_{i,t} = \beta_1 \ln(\text{GDP}_{pc})_{i,t} + \beta_2 \text{education}_{i,t} + \beta_3 \text{educationgap}_{i,t} + \alpha_i + \varepsilon_{i,t} \quad (1)$$

where $\alpha_i$ captures country fixed effects and $\varepsilon_{i,t}$ is the error term, assumed to be stationary. Several robustness checks carried out by changing the specification can be found in Supplementary Table 1.

Projections for the 21st century are carried out by combining the parameter estimates from the specification given by Eq. (1) with the existing projections of GDP[34], population by age, sex and education[36] and gender gap in education[36] thereby remaining internally consistent with the SSP scenario framework and providing direct comparability with the rest of the socioeconomic projections existing. The SSP population projections[36] were employed to derive the proportion of women experiencing different levels of gender inequality in the future at the global level. We split the population of women into two age groups: 0–14 and 15+. The thresholds for dividing the distribution of GII are based on the levels of gender inequality currently in the OECD countries (0.002–0.315).

**Reporting summary**. Further information on research design is available in the Nature Research Reporting Summary linked to this article.

## Data availability

Original GII data is available through the UNDP website (http://hdr.undp.org/en/data). Data on maternal mortality ratio is available from UNICEF (https://data.unicef.org/topic/maternal-health/maternal-mortality/), and adolescent birth rates from WHO (https://www.who.int/gho/maternal_health/reproductive_health/adolescent_fertility/en/). Historical GDP was obtained from the Penn World Tables 7.0 (https://www.rug.nl/ggdc/productivity/pwt/pwt-releases/pwt-7.0) and projected values through the IIASA SSP database (https://tntcat.iiasa.ac.at/SspDb/). Data on educational attainment and gender gap in mean years of schooling is accessible through the Data Explorer of the Wittgenstein Centre for Demography and Global Human Capital (http://dataexplorer.wittgensteincentre.org/wcde-v2/).

## Code availability

Code underlying the results is available at https://github.com/marina-andrijevic/gender_equality2020.

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

## Acknowledgements

The authors express their gratitude to the scientific community for developing the SSP scenarios and to the International Institute for Advanced System Analysis (IIASA) for hosting the SSP database. M.A. and C.F.S. acknowledge support by the German Federal Ministry of Education and Research (01LN1711A).

## Author contributions

The research was designed by M.A. and C.F.S. M.A. and J.C.C. performed the analysis and M.A. created the display items. M.A., J.C.C., T.L., A.T., and C.F.S. contributed to the writing of the manuscript.

## Funding

## Competing interests

The authors declare no competing interests.
