## [Peer Review File · Nature Communications]

Reviewers' comments:

Reviewer #1 (Remarks to the Author):

This paper investigates trajectories of the UNDP's Gender Inequality Index (GII) alongside the different Shared Socio-economic Pathways (SSPs). A priori, this looks like an interesting exercise. Gender inequality is a very important dimension that should be incorporated when evaluating societies' overall well-being levels, so there seem to be good reasons to look at its evolution alongside other key socio-economic indicators within the SSPs framework.

Yet, I am not fully convinced about the usefulness of such exercise. I understand that SSPs scenarios quantify different narratives of socio-economic futures, so they are speculative in nature, and that the projections of the GII should not be interpreted as predictions, but as an attempt to quantify narrative-driven scenarios. However, to speculate about the trajectories of a set of socio-economic variables X is one thing, but to further speculate about the trajectory of a new variable Y that depends in a non-trivial way on X is another thing. This double speculation exercise crucially hinges on equation (1) – a simple model that links the values of (a logit transformation of) the GII with several socio-economic indicators – and on the extremely strong assumption that such relationship will hold during the next 80 years. Unfortunately, nothing is known about the goodness of fit of equation (1). Has this model been tested for the previous years with observed data (e.g. from 2000 to 2020)? How good is it at predicting GII values? And more importantly: has the goodness of fit changed over time? The answer to these questions would lend support to (or eventually reject) the hypothesis that the relationship shown in equation (1) can be meaningfully projected almost one century into the future. If the paper were invited for resubmission, these important issues should be addressed.

Lastly, while the GII incorporates interesting variables to assess the levels of gender inequality, it has been widely criticized for its conceptual and methodological flaws. On the one hand, it incorporates indicators where the performance of women is compared vis-à-vis the performance of men (e.g. the education or labor force participation indicators). On the other hand, it incorporates women-specific indicators (maternal mortality ratio and adolescent birth rate) with no male counterpart. The result is an odd indicator without a clear interpretation that unduly penalizes low-income countries for scoring badly in the reproductive health dimension, but not for the gender differences in health that the indicator is purported to measure.

Reviewer #2 (Remarks to the Author):

The purpose of this paper is to understand the role of gender inequality as it intersects with vulnerability and adaptation to climate change projections. This is an important topic, but I do not feel the paper at present is in publishable form. It is unclear why certain measures are used and what the major conclusions/implications are. I provide further details of my assessment in my comments below.

First, the authors should provide more information on the SSPs used in the paper. I would like more information on how this is measured, past findings, and limitations of the approach/measure. E.g.: Who has used the toolkit? What sort of applications exist? Why not cite them?

Similarly, the authors use GII to operationalize gender inequality, but more information on this measure (and other alternatives) is needed. Why did the authors choose to employ the GII? How does it compare/contrast to other possible indicators on gender equality? Why is GII preferable? The authors should explain if GII is based solely on data for women (with no basis of comparison to men e.g., secondary school enrollments) or relative (to men) measures. Some indicators (e.g., maternal mortality ratio) used to compile the GII index have no basis for comparison to men, whereas others (e.g., secondary school enrollments, participation in politics & labor) do have

baseline data for comparisons. Which are used in formulating the GII? For readers who are less familiar with the GII, the authors should explain if high numbers indicate greater inequality or less inequality. These details matter. In sum, the authors would do well to explain the nuances of the data and justify their reasons for choosing GII in the analysis.

Related to this, the authors state on pg. 91 (pg. 5 of PDF) that GII is favorable to developed countries, and the GDI and GEM tend to penalize low-income countries (which is another way of saying they, too, favor developed countries). They then offer that GII a "significant improvement" compared to GDI and GEM, but how is this so when GII suffers from the same tendency of GDI and GEM data to favor developed countries?

Figure 1a correlates GII with the Notre Dame Global Adaptation Index (ND-GAIN); Figure 1b correlates GII with the Climate Laws, Institutions, and Measure Index (CLIMI). I have the same questions for these choices: how are the indexes measured/operationalized? Have others used them? If so, how and what did they find? Why do you opt for these measures compared to other possibilities? Also, are the correlations as you would expect based on prior analyses?

Figure 3: I only see four lines (scenarios) in the figures. One line is missing: I believe the line for SSP5 is missing.

Figure 4: It is unclear to me why projections for SSP 4 & SSP5 are omitted.

Manuscript NCOMMS-20-14444-T

Overcoming gender inequality for climate resilient development

Andrijevic M., Crespo Cuaresma J., Lissner T., Thomas A., Schleussner C.-F.

Response to reviewers

We would like to thank both reviewers for their insightful comments and for demanding further reflection on the implications of our work and the use of the particular indicator of gender inequality. The manuscript benefited substantially as a result of the reviewers' remarks and we thank them for reconsideration of our paper. Below we address individual points.

Reviewer #1

1.1. This paper investigates trajectories of the UNDP's Gender Inequality Index (GII) alongside the different Shared Socio-economic Pathways (SSPs). A priori, this looks like an interesting exercise. Gender inequality is a very important dimension that should be incorporated when evaluating societies' overall well-being levels, so there seem to be good reasons to look at its evolution alongside other key socio-economic indicators within the SSPs framework.

We thank the reviewer for the positive assessment of the motivation behind this paper. We indeed agree that gender inequality is a key dimension of overall societal well-being, and as such is indispensable for analyses of future socio-economic development and capacity to adapt to climate change.

1.2. Yet, I am not fully convinced about the usefulness of such exercise. I understand that SSPs scenarios quantify different narratives of socio-economic futures, so they are speculative in nature, and that the projections of the GII should not be interpreted as predictions, but as an attempt to quantify narrative-driven scenarios. However, to speculate about the trajectories of a set of socio-economic variables X is one thing, but to further speculate about the trajectory of a new variable Y that depends in a non-trivial way on X is another thing. This double speculation exercise crucially hinges on equation (1) – a simple model that links the values of (a logit transformation of) the GII with several socio-economic indicators – and on the extremely strong assumption that such

relationship will hold during the next 80 years. Unfortunately, nothing is known about the goodness of fit of equation (1).

Has this model been tested for the previous years with observed data (e.g. from 2000 to 2020)? How good is it at predicting GII values? And more importantly: has the goodness of fit changed over time? The answer to these questions would lend support to (or eventually reject) the hypothesis that the relationship shown in equation (1) can be meaningfully projected almost one century into the future. If the paper were invited for resubmission, these important issues should be addressed.

We thank the reviewer for raising these important concerns. We have now expanded the results reported in the Supplementary Material to show measures of goodness-of-fit beyond R-squared, and have performed additional robustness checks.

Specifically, we assess the predictive ability of the variables used and the model employed using a validation exercise based on an out-of-sample predictive exercise. Using data spanning the period 2000-2005, we estimate an autoregressive model for our gender inequality variable, which serves as a benchmark to evaluate the (out-of-sample) predictive content of the information contained in the covariates of our specification. The autoregressive specification is given by

$$GII_{i,t}^* = \alpha_i + \vartheta GII_{i,t-5}^* + \varepsilon_{i,t},$$

implying that the dynamics of the gender inequality index can be explained by mean reverting dynamics around a country-specific equilibrium which is given by $\alpha_i/(1 - \vartheta)$. Using this specification after estimating it for the period 2000-2005, we can obtain out-of-sample forecasts for all the countries in our sample for the year 2010. We also estimate a model that includes information about GDP per capita, education and the education gap, the three driving factors of gender inequality we consider in our main specification,

$$GII_{i,t}^* = \alpha_i + \vartheta GII_{i,t-5}^* + \beta_1 \ln GDPpc_{i,t-5} + \beta_2 education_{i,t-5} + \beta_3 educationgap_{i,t-5} + \varepsilon_{i,t},$$

where the covariates enter with a lag of five years to allow for five years-ahead out-of-sample predictions. After estimating this specification for the period 2000-2005, we can obtain predictions of the gender inequality index in 2010 for the countries in our sample based on a model that includes information on income and education dynamics. Expanding the set of in-sample observations to 2000-2006, we can obtain out-of-sample predictions for the year 2011, and repeating this exercise by expanding the sample used to estimate the model we can obtain 1202 five years-ahead forecasts spanning the period 2010-2017.

Table 2 in the Supplementary Material presents several standard measures of predictive error for the autoregressive (AR) specification and our model (MODEL) based on these forecasts. We compute (i) the mean squared forecast error (MSFE), which is the average of the squared deviations between realized and forecast values; (ii) the directional accuracy (DA) statistic, which gives the percentage of out-of-sample observations whose direction of change (increase or decrease) was correctly predicted, and (iii) the directional value (DV), which gives the average absolute value of the correctly predicted changes and should inform about whether the corresponding model fails at forecasting important changes in the target variable.

	AR	MODEL
RMSFE	0.306	0.283
DA	56.32%	68.64%
DV	0.152	0.207
Obs.	1202	1202

Supplementary Table 2: Model vs. benchmark AR specification

The results of the validation exercise based on the out-of-sample predictive ability of the model used give clear evidence that the covariates used in the model contain predictive information about future changes in the Gender Inequality Index. In addition to reducing MSFE, the use of variables related to income, education and its distribution across genders increases directional accuracy very substantially, from around 56% correctly predicted changes to almost 69%. In addition, the changes which are forecast correctly are on average larger than those in the benchmark specification.

1.3. Lastly, while the GII incorporates interesting variables to assess the levels of gender inequality, it has been widely criticized for its conceptual and methodological flaws. On the one hand, it incorporates indicators where the performance of women is compared vis-à-vis the performance of men (e.g. the education or labor force participation indicators). On the other hand, it incorporates women-specific indicators (maternal mortality ratio and adolescent birth rate) with no male counterpart. The result is an odd indicator without a clear interpretation that unduly penalizes low-income countries for scoring badly in the reproductive health dimension, but not for the gender differences in health that the indicator is purported to measure.

We agree that the health dimension of the GII does not intuitively align with the other two dimensions of the index, as it contains female-specific measures. However, poor reproductive health or adolescent pregnancy can disadvantage women's health and future prospects in many different ways, from health risks to dropping out of school and the labor market. These potential setbacks therefore uniquely affect women, and while they do not have a direct equivalent for men, such setbacks reflect preventable disparities in health. Adolescent birth rate and maternal mortality are much more drastic indicators for health inequalities arising from the state of reproductive health and, from a capabilities standpoint, their inclusion can be justified. Not least, universal access and sexual and reproductive health and reproductive rights are included in the gender equality goal (SDG5.6) as well as in the global health goal (SDG3.1).

We fully take the reviewer's point with regard to the critical reflection of whether or not this results in a penalization of low-income countries with a generally weaker health system or preventive public health campaigns. However, we also acknowledge that the resulting health perils are gender-specific - which is what this indicator is capturing. We also note that the relationship between economic development and reproductive health indicators is not entirely straightforward (for example, the rate of adolescent pregnancy of the US is just about the same as Iran's and more closely matched by India than by EU countries¹).

Additionally, it is important to note that the primary intention of this index being better understanding of the capacity to adapt to climate change, for which it is relevant to point out a strand of research showing that reproductive health is strongly affected by climatic changes (see, for example, references^{2,3}).

Reviewer #2:

2.1. The purpose of this paper is to understand the role of gender inequality as it intersects with vulnerability and adaptation to climate change projections. This is an important topic, but I do not feel the paper at present is in publishable form. It is unclear why certain measures are used and what the major conclusions/implications are. I provide further details of my assessment in my comments below.

We thank the reviewer for acknowledging the importance of the topic and drawing our attention to the shortcomings in delivering our message. We have substantially revised the paper to clarify both the purpose and the conclusions of this work. Below we address each individual point.

2.2. First, the authors should provide more information on the SSPs used in the paper. I would like more information on how this is measured, past findings, and limitations of the approach/measure. E.g.: Who has used the toolkit? What sort of applications exist? Why not cite them?

We thank the reviewer for pointing out the lack of elaboration of the SSP scenarios. We note that the SSP framework is a central pillar of climate science research widely used in reports of the Intergovernmental Panel on Climate Change (IPCC) and more than 10.000 publications (based on the search of the Web of Science). We fully acknowledge the need to provide additional detail in the SSPs to also familiarise readers from other disciplines with the concept of those pathways and to provide authoritative referencing for further reading. We have also outlined more clearly, what variables are originally quantified as part of the SSP storylines and how our extensions relate to those dimensions.

“The SSPs are scenarios that explore a range of possible futures that illustrate how socio-economic conditions might change over the next century and what implications these conditions may have for climate change adaptation and mitigation. SSPs quantify five different narratives of socio-economic futures to operationalize them for climate change research¹¹ – they are a widely used tool in climate research community, indispensable for integrated assessments of the dynamics between socioeconomic and climate change variables, and are also the scenario framework used in the Sixth Assessment report of the IPCC.

SSP1, the ‘sustainability’ scenario, is characterized by low challenges to mitigation and adaptation, a result of increased investments in education, health, renewable energy sources and declining inequalities between and within countries, thus limiting impacts and increasing adaptive capacity. SSP2, the ‘middle of the road’ scenario, maintains premediated challenges to adaptation and mitigation, and is a pathway of uneven and slower socioeconomic progress, compatible with the continuation of historical trends. SSP3 is characterized by high challenges to both mitigation and adaptation, which are a product of a growing divergence between economies, weak international cooperation and increase in internal and international conflicts. SSP4, the scenario of ‘inequality’, leads to low challenges for mitigation, due to technological advancements in high income countries, but high challenges for adaptation, because of an unequal distribution of advancements and resources across countries. Finally, SSP5 is similar to SSP1 in the fast socioeconomic progress on all fronts, but with the major difference of the progress being powered by fossil fuels, which produces substantially higher emissions and resulting climate impacts.

So far, the SSPs storylines have been quantified in future trajectories of income^{33,34}, population³⁵, education³⁵, urbanization³⁶, the Human Development Index³⁷, inequality³⁸ and governance³⁹. Gender inequality is qualitatively featured in the scenarios' storylines focusing on the demographic and human development elements (see Table 1), and is to a certain extent reflected in the measures of discrepancies in educational attainment between men and women in the population projections by age and sex³⁵. Our contribution provides projections of gender inequality, as quantified by the GII, which are compatible with the SSP scenarios described above and thus provide a new dimension to the assessment of potential future climate change adaptation pathways."

2.3. Similarly, the authors use GII to operationalize gender inequality, but more information on this measure (and other alternatives) is needed. Why did the authors choose to employ the GII? How does it compare/contrast to other possible indicators on gender equality? Why is GII preferable?

We thank the reviewer for pointing out the need for justification of our use of GII and the comparison to other indicators. In our original manuscript, we have attempted to do so in the Methods part of our manuscript providing a detailed explanation of the GII, as well as comparisons with three other indicators of gender (in)equality.

Encouraged by the reviewer's comment we have now changed the structure of the main text to dedicate an entire section to elaborate on indicators of gender inequality and expand the description of the GII.

"The GII used here to indicate gender inequality consists of three dimensions: health (maternal mortality ratio and adolescent birth rates), educational and political empowerment (male to female ratio in parliamentary seats and secondary education) and participation in the labor market (male to female ratio in labor force participation rates, see the Methods section for additional details on the indicator)¹². We collected the individual components from their respective original sources and reconstructed the index following the approach laid out in the Technical Notes of the Human Development Report¹². This reconstruction produced more complete time series than those available hitherto (see Supplementary Fig. 1). The index ranges from 0 to 1, with higher values reflecting higher levels of inequality between men and women.

Quantifying gender inequality remains a daunting task. The multi-faceted nature of gender inequality on all levels on socio-economic development makes aggregation into indicator a complex exercise. Unsurprisingly, most indicators including the GII, face (well-justified) criticism^{13,14}. The GII used can be seen to

be unfavorable towards low developed countries, as a result of the indicator's underlying dimensions such as maternal mortality and adolescent birth rates. This health dimension of the GII considers variables that do not have a male equivalent, unlike the dimensions of economic, political and labor market metrics. The rationale behind accounting for maternal mortality and adolescent birth rate as a dimension of a gendered health inequality stems from the fact that maternal health sets women back uniquely, without an equivalent metric for men, and as such arguably contributes to gender inequality. Reducing maternal mortality and adolescent pregnancy are also among the targets of the Sustainable Development Goals (SDGs, Goal 3 and 5). Additionally, reproductive health is strongly affected by climate change impacts such as extreme heat, and as such merits consideration as an own standing dimension of climate adaptation¹⁵.

Compared to other commonly used indicators such as the Gender Development Index¹², the Gender Empowerment Measure¹⁶, and the Women, Peace and Security Index¹⁷, we find that the GII is particularly indicative of hindered adaptive capacity in many climate-vulnerable countries, since its dimensions (such as maternal health, participation in economic and political life) point at the very basic disempowerment of women that directly reduces their capacity to adapt to climate change. The GII is also more holistic in its economic dimension, by considering education and labor force participation rather than income, since the data on gender gap in earned income tends to be problematic¹⁸. In addition, the construction of the GII precludes the different dimensions of the indicator from compensating for each other (i.e. poor performance in one dimension can be compensated for with higher performance in another dimension). For a more in-depth qualitative and quantitative comparison, see the Methods section.

We consider the dimensions covered in the GII to describe necessary conditions of gender inequality, while acknowledging that they are not sufficient to characterize gender inequality across all the dimensions that contribute to it. In the light of these caveats, overcoming the inequality dimensions covered in the GII does not automatically mean that universal gender equality is achieved, and we do not assert that any country in the world can claim to have achieved full gender equality to date or in the near future. It is important to keep these limitations in mind when interpreting the results."

2.4. The authors should explain if GII is based solely on data for women (with no basis of comparison to men e.g., secondary school enrollments) or relative (to men) measures. Some indicators (e.g., maternal mortality ratio) used to compile the GII index have no basis for comparison to men, whereas others (e.g., secondary school enrollments, participation in politics & labor) do have baseline data for comparisons. Which are used in formulating the GII? For readers who are less familiar with the GII, the authors should explain if high numbers indicate greater inequality or less inequality.

These details matter. In sum, the authors would do well to explain the nuances of the data and justify their reasons for choosing GII in the analysis.

We thank the reviewer for referring to the lack of clarity on this aspect of the index - the manuscript has now been edited to better describe the three dimensions of the indicator, as well as the range. The health dimension indeed does not have an intuitive male counterpart (see also our response to comment 1.3 of Reviewer #1 above) . However, poor reproductive health or adolescent pregnancy can affect women in many different ways, from greater health risks to dropping out of school and the labor market. These potential setbacks therefore uniquely affect women, and while they do not have a direct equivalent for men, such setbacks reflect preventable disparities in health.

Together with addressing the comment 2.3, we believe that the case for using the GII in this analysis has been made stronger. We have extended the respective sections of the paper.

2.5. Related to this, the authors state on pg. 91 (pg. 5 of PDF) that GII is favorable to developed countries, and the GDI and GEM tend to penalize low-income countries (which is another way of saying they, too, favor developed countries). They then offer that GII a “significant improvement” compared to GDI and GEM, but how is this so when GII suffers from the same tendency of GDI and GEM data to favor developed countries?

We thank the reviewer for pointing out this inconsistency in the narrative, which we have now clarified. The advantage of the GII over the other widespread indicators of gender (in)equality has been misspecified in the earlier version of the manuscript, and now instead reads:

“Compared to other commonly used indicators such as the Gender Development Index¹², the Gender Empowerment Measure¹⁶, and the Women, Peace and Security Index¹⁷, we find that the GII is particularly indicative of hindered adaptive capacity in many climate-vulnerable countries, since its dimensions (such as maternal health, participation in economic and political life) point at the very basic disempowerment of women that directly reduces their capacity to adapt to climate change. The GII is also more holistic in its economic dimension, by considering education and labor force participation rather than income, since the data on gender gap in earned income tends to be problematic¹⁸. In addition, the construction of the GII precludes the different dimensions of the indicator from compensating for each other (i.e. poor performance in one dimension can be

compensated for with higher performance in another dimension). For a more in-depth qualitative and quantitative comparison, see the Methods section.”

We furthermore note the link between the GII and indicators underlying the SDGs (in particular SDG 3 and 5).

2.6. Figure 1a correlates GII with the Notre Dame Global Adaptation Index (ND-GAIN); Figure 1b correlates GII with the Climate Laws, Institutions, and Measure Index (CLIMI). I have the same questions for these choices: how are the indexes measured/operationalized? Have others used them? If so, how and what did they find? Why do you opt for these measures compared to other possibilities? Also, are the correlations as you would expect based on prior analyses?

We have now supplemented the main text with more detailed explanation of the ND-GAIN index and the CLIMI. We motivate our main contribution (the projections of GII) by correlating the indicator gender inequality with the two already established indicators relevant for climate change adaptation (ND-GAIN, see more applications in references⁴⁻⁶) and mitigation measures (CLIMI, see more applications in references⁷⁻⁹), we provide quantitative backing for previous research findings on the gendered nature of vulnerability and case studies reporting that women tend to be disproportionately affected by climate change. Additionally, we refer to the previous research on the relationship between female empowerment in politics and climate action, and correlate our index of gender inequality to the index of climate action and similarly find that higher levels of gender inequality correlate with weaker climate action.

2.7. Figure 3: I only see four lines (scenarios) in the figures. One line is missing: I believe the line for SSP5 is missing.

We thank the reviewer for pointing out the lack of explanation for the overlap between scenarios SSP1 and SSP5 in the case of GII. The two scenarios are similar in the estimates of the underlying dimensions that the GII is a function of (namely education, GDP and gender gap in mean years of schooling). The marked difference between the two scenarios in the nature of their economic growth (fossil fuel-intensive vs. low carbon) and the greenhouse gas emissions that they produce, which is not relevant for this analysis. Since their trajectories are similar, gender inequality estimates are also consistently similar, which is the reason for the two lines almost entirely overlapping. We have clarified this in the text to avoid confusion.

“Note that the trajectories for SSPs 1 and 5 largely overlap due to similar levels of the underlying dimensions that gender inequality is a function of (education, GDP and gender gap in mean years of schooling).”

2.8. Figure 4: It is unclear to me why projections for SSP 4 & SSP5 are omitted.

We show only SSPs 1,2 and 3 for reasons of brevity. Those three scenarios display the largest differences between them and span the full range of the scenario space. SSP4 is in this case rather similar to SSP3, and SSP1 and SSP5 largely overlap (as clarified in the response to the point 2.8). We have now clarified the reasons for omitting the two scenarios, and thank the reviewer for pointing out the lack of clarity.

“Note that for reasons of brevity we here show only scenarios 1-3, which encompass the full range of the five scenarios, and exhibit large differences between each other.”

References:

1. Adolescent birth rate (per 1000 women aged 15-19 years). Available at: <https://www.who.int/data/gho/indicator-metadata-registry/imr-details/4669>. (Accessed: 29th June 2020)
2. Watt, S. & Chamberlain, J. Water, climate change, and maternal and newborn health. *Current Opinion in Environmental Sustainability* **3**, 491–496 (2011).
3. Bekkar, B., Pacheco, S., Basu, R. & Denicola, N. Association of Air Pollution and Heat Exposure With Preterm Birth , Low Birth Weight , and Stillbirth in the US A Systematic Review. **3**, 1–13 (2020).
4. Adams, C., Ide, T., Barnett, J. & Detges, A. Sampling bias in climate-conflict research. *Nat. Clim. Chang.* **8**, 200–203 (2018).
5. Robinson, S.-A. & Doman, M. International financing for climate change adaptation in small island developing states. *Reg. Environ. Chang.* **17**,
6. Lesnikowski, A. C. *et al.* How are we adapting to climate change? A global assessment. *Mitig Adapt Strateg Glob Chang.* **20**, 277–293 (2015).
7. Mavisakalyan, A. & Tarverdi, Y. Gender and climate change: Do female parliamentarians make difference? *Eur. J. Polit. Econ.* **56**, 151–164 (2019).
8. OECD. Environmental Policy Stringency index. *OECD Environ. Stat.* (2017). doi:10.1787/5JXRJNC45GVG-EN
9. Fredriksson, P. G. & Neumayer, E. Democracy and climate change policies: Is history important? *Ecol. Econ.* **95**, 11–19 (2013).

REVIEWERS' COMMENTS

Reviewer #1 (Remarks to the Author):

I think the authors have satisfactorily addressed the technical issues raised in my previous report (i.e. regarding the quality and accuracy of the estimation model). Yet, I remain somewhat skeptical about the substantive meaning of the exercise of projecting the values of a flawed index. I understand that all composite indices have their limitations and drawbacks (i.e. there is no "perfect" index), and I also understand that the Gender Inequality Index (GII) incorporates important indicators of reproductive health that are crucial for women's well-being. However, the way in which these important variables are combined (i.e. the functional form of the index) generates an artificially complex and strange composite indicator whose values are very difficult to understand. The GII has been a useful tool to raise awareness among the general public about reproductive health problems affecting women, particularly in low-income settings. Yet, from a scientific perspective, the index has been severely criticized on several grounds by renowned experts in the field (e.g. Klasen & Schüler 2011, Permanyer 2013, Klasen 2018) – who suggest dropping the GII methodology altogether. As a reaction to such criticism, sooner rather than later the UNDP intends to release new measures of gender inequality that overcome the limitations of the GII.

References

Klasen, S. and Schüler, D. (2011), "Reforming the Gender-Related Development Index and the Gender Empowerment Measure: Implementing Some Specific Proposals" *Feminist Economics* 17(1):1-30.

Klasen, S. (2018), "Human Development Indices and Indicators: A critical evaluation", Human Development Report Office Background Paper.

Permanyer, I. (2013), "A Critical Assessment of the UNDP's Gender Inequality Index", *Feminist Economics* 19(2):1-32.

Reviewer #2 (Remarks to the Author):

The authors have done an excellent job attending to the comments provided.

Overcoming gender inequality for climate resilient development

NCOMMS-20-14444A-Z

Response to reviewers

Reviewer #1:

I think the authors have satisfactorily addressed the technical issues raised in my previous report (i.e. regarding the quality and accuracy of the estimation model). Yet, I remain somewhat skeptical about the substantive meaning of the exercise of projecting the values of a flawed index. I understand that all composite indices have their limitations and drawbacks (i.e. there is no “perfect” index), and I also understand that the Gender Inequality Index (GII) incorporates important indicators of reproductive health that are crucial for women’s well-being. However, the way in which these important variables are combined (i.e. the functional form of the index) generates an artificially complex and strange composite indicator whose values are very difficult to understand. The GII has been a useful tool to raise awareness among the general public about reproductive health problems affecting women, particularly in low-income settings. Yet, from a scientific perspective, the index has been severely criticized on several grounds by renowned experts in the field (e.g. Klasen & Schüler 2011, Permanyer 2013, Klasen 2018) – who suggest dropping the GII methodology altogether. As a reaction to such criticism, sooner rather than later the UNDP intends to release new measures of gender inequality that overcome the limitations of the GII.

Klasen, S. and Schüler, D. (2011), “Reforming the Gender-Related Development Index and the Gender Empowerment Measure: Implementing Some Specific Proposals” *Feminist Economics* 17(1):1-30.

Klasen, S. (2018), “Human Development Indices and Indicators: A critical evaluation”, Human Development Report Office Background Paper.

Permanyer, I. (2013), “A Critical Assessment of the UNDP’s Gender Inequality Index”, *Feminist Economics* 19(2):1-32.

Authors’ response:

We were glad to learn that we addressed the technical issues raised by the reviewer, and are thankful for the additional reflections in the new report. We would like to use this opportunity to substantiate our methodological decisions and further clarify the motivation behind this manuscript. In line with this response, we have extended the discussion on the GII indicator in the Data section of the manuscript (lines 448-473; 484-486).

We acknowledge that the criticism brought forward primarily by Klasen and Permanyer does raise important issues with the GII index, most notably regarding its functional form (which is asserted to be unnecessarily complex and challenging to interpret) and its composition (i.e. accounting for women-specific dimensions without an adequate counterpart for men, as well as the penalization of low-income countries where poverty rather than inequality might be the cause of the poor state of reproductive health care). Since no satisfactory consensus has been achieved among the critics on the particular nature of an optimal index, we decided to stick with what was legitimized by the UNDP and what provides the most relevant reference for the policy discussion. In addition, we are reassured by Permanyer’s (2013) finding that the simplified Woman Disadvantage (WD) index (which does not include a health dimension) proposed in his study produces results analogous to the capped GII (GII without the reproductive health component). Still, it remains unclear what health dimension should the improved index account for instead.

As for Klasen’s (2018) assertion regarding penalization of low-income countries, where we also see space for improvement methodologically, we are of the opinion that poor maternal health can

contribute to gender inequality, rather than merely indicate it, because it sets women back in unique ways that do not have the male counterpart. From a slightly different angle, we find support for the GII in recent applications where it was found to explain variance in child malnutrition and mortality in low and middle income countries with similar income levels, suggesting the correlation of the index with maternal health, independently of GDP (Brinda, 2015; Marphatia 2016).

We also find support for the GII in research findings that indicate that the GII correlates with other dimensions of gender inequality (which go beyond what is accounted in the GII or proposed alternatives), such as the suicide gender ratio (Chang et al., 2019), adolescent dating violence (Gressard et al., 2015) and intimate partner violence (Redding et al., 2017).

While we agree in principle with the shortcomings identified in the existing literature, the intention of our study is not to contribute to the discussion on the particular advantages and disadvantages of the GII index. Despite the criticism, the GII is still routinely used by national and international institutions in policy discussions. Since the audiences we intend to reach with this work - mostly researchers and policy analysts in the arena of climate change adaptation and mitigation - are expected to be familiar with this and other indices by the UNDP that are prominently featured in the nexus of socio-economics and climate change, the proposal of a methodologically improved index would not contribute to the aim of our study.

Although the methodological improvements of aggregated gender inequality indicators is beyond the scope of our work, we encourage future research to apply the analytical framework that we propose here to different versions of gender inequality indicators, as well as potential revisions and advances from the UNDP's indices. All of the proposals to reform the index can be implemented in our methodological framework by applying it to particular components of the GII (or another gender inequality index). This would allow for the computation of projections that can then be aggregated to give rise to different indices. We explicitly recommend this as a potentially fruitful path of further research in the revised paper. Since our work is pioneering in terms of accounting for gender inequality in quantitative research on climate change impacts and adaptation, we are convinced that it remains a valuable contribution for the broad readership of the journal in its present form.

References:

1. Brinda, E. M., Rajkumar, A. P., & Enemark, U. (2015). Association between gender inequality index and child mortality rates: a cross-national study of 138 countries. *BMC public health*, 15(1), 97.
2. Chang, Q., Yip, P. S., & Chen, Y. Y. (2019). Gender inequality and suicide gender ratios in the world. *Journal of affective disorders*, 243, 297-304.
3. Gressard, L. A., Swahn, M. H., & Tharp, A. T. (2015). A first look at gender inequality as a societal risk factor for dating violence. *American journal of preventive medicine*, 49(3), 448-457.
4. Klasen, S. (2017). *UNDP's gender-related measures: Current problems and proposals for fixing them* (No. 220). Discussion Papers.
5. Marphatia, A. A., Cole, T. J., Grijalva-Eternod, C., & Wells, J. C. (2016). Associations of gender inequality with child malnutrition and mortality across 96 countries. *Global health, epidemiology and genomics*, 1.
6. Permanyer, I., 2013. A critical assessment of the UNDP's gender inequality index. *Feminist Economics*, 19(2), pp.1-32.
7. Redding, E. M., Ruiz-Cantero, M. T., Fernández-Sáez, J., & Guijarro-Garvi, M. (2017). Gender inequality and violence against women in Spain, 2006-2014: towards a civilized society. *Gaceta sanitaria*, 31, 82-88.

Reviewer #2:

The authors have done an excellent job attending to the comments provided.

Authors' response:

We thank the reviewer for the positive feedback and useful comments in the first round of the review.